# HyperPINN: Learning parameterized differential equations with physics-informed hypernetworks

**Filipe de Avila Belbute-Peres**[*]
School of Computer Science
Carnegie Mellon University
filiped@cs.cmu.edu

**Fei Sha**
Google Research
fsha@google.com

**Yi-fan Chen**
Google Research
yifanchen@google.com

## Abstract

Many types of physics-informed neural network models have been proposed in recent years as approaches for learning solutions to differential equations. When a particular task requires solving a differential equation at multiple parameterizations, this requires either re-training the model, or expanding its representation capacity to include the parameterization – both solution that increase its computational cost. We propose the HyperPINN, which uses hypernetworks to learn to generate neural networks that can solve a differential equation from a given parameterization. We demonstrate with experiments on both a PDE and an ODE that this type of model can lead to neural network solutions to differential equations that maintain a small size, even when learning a family of solutions over a parameter space.

## 1 Introduction

The recent successes of deep learning approaches in diverse domains have motivated many works exploring their application to physical systems, including approximating the solutions to differential equations [2, 3, 6, 7, 9, 11]. In many applications such as shape optimization, topology optimization or design prototyping, approximate solutions with fast iteration times might be preferred over guaranteed accurate, yet computationally complex, ones. In these cases, machine learning models might offer an interesting alternative to traditional methods. Moreover, the usage of data-driven methods allows for the incorporation of data into the solution, which can be useful in domains where only noisy or partial measurements are available, or where the underlying physics are not fully known [4, 10, 12].

Physics-informed neural networks (PINNs) [9] have been recently proposed as a method for employing neural networks as function approximators to the solution of differential equations, while allowing the incorporation of the underlying physical knowledge in the form of a physics-informed loss. In their standard formulation, PINNs fit the solution to a single parameterization of a differential equation. Thus, when working in a domain that requires evaluating the solutions at multiple parameterizations, this requires either (1) re-training the model multiple times to find the solution at each parameterization, or (2) including the parameterization explicitly as an input to the neural network model (e.g., [1]). Given that training is the most expensive part of the process, having to repeat the learning procedure for every parameterization can greatly increase the computational cost of the method. Conversely, augmenting the model to take into account the parameters requires increasing

---

[*]Work done while at Google Research.

35th Conference on Neural Information Processing Systems (NeurIPS 2021), Sydney, Australia.

the capacity of the neural network, as it now has to both approximate the solution function *and* model the parameter space, thus also increasing the computational cost at inference time.

In this work, we propose looking at the problem of learning parameterized families of differential equations as a meta-learning problem, where the given parameters and initial/boundary conditions define a task, which can then be solved by a neural network. Under this framework, we propose the HyperPINN, which uses a hypernetwork [5] to learn to model the parameter space of a differential equation, taking as input a given parameterization and producing as output a main network that approximates the solution function at that specific parameterization. By separating this task into two parts, the complexity of modeling the parameter space is "offloaded" to the hypernetwork, which is only evaluated once for every parameterization. Importantly, this allows the main network, which is evaluated at every time-space point, to remain small. We demonstrate this approach with experiments on a PDE and an ODE, using both "standard" PINNs [9] and multistep neural networks [8].

## 2   Methods

Let us assume a differential equation in the general form $\mathcal{N}[t, x, u(t, x); \lambda] = 0$, defined by the (possibly non-linear) differential operator $\mathcal{N}[\cdot; \lambda]$, which can contain time and space derivatives, and is parameterized by some list of parameters $\lambda \in \mathbb{R}^d$. Refer to Appendix A.1 for a complete definition.

**Physics-Informed Neural Networks**   Physics-informed neural networks [9] are a method for approximating the solution to differential equations using neural networks (NNs). In this method, a neural network $\hat{u}(t, x; \theta)$, with learned parameters $\theta$, is trained to approximate the solution function $u$ to the differential equations. Importantly, PINNs employ not only a standard "supervised" data loss, but also a physics-informed loss, which consists of the differential equation residual defined by $\mathcal{N}$. Refer to Appendix A.2 for further details.

**Multistep Neural Networks**   Multistep neural networks [8] are a related method, in which a neural network model is used to approximate the time-derivative of a dynamical system. Instead of using the differential equation residiuals, the loss for the multistep neural network is derived directly from the formulation for traditional multistep methods. Refer to Appendix A.3 for further details.

**Hypernetworks**   Hypernetworks [5] are a recently proposed meta-learning method in which learning is broken down into two separate networks: a main network and a hypernetwork. The main network performs the desired task, in the same way a neural network would normally be employed. The parameters for this network, however, are not learned directly at training time. Instead, the parameters for the main network are generated, at evaluation time, by the hypernetwork. That is, for a hypernetwork $f_h$ that takes an input $x_h$, and a main network $f_m$ that takes as input $x_m$, we have

$$\theta_m = f_h(x_h; \theta_h), \ \hat{y} = f_m(x_m; \theta_m), \tag{1}$$

where $\theta_h$ are the learnable parameters. This ability to generate neural networks allows the hypernet to meta-learn a space of tasks defined by $x_h$, outputting an appropriate main network for each task.

**HyperPINN**   In this work, we propose the HyperPINN, which combines the previously described physics-informed architectures with hypernetworks in order to learn parameterized families of differential equations. A HyperPINN has a hypernetwork that, given a certain parameterization, generates a main network that approximate the solution to the corresponding differential equation.

Following the general differential equation definition above, we want to have a hypernetwork that maps a parameterization $\lambda$ into an approximation of the differential equation given by a neural network. Following the hypernetwork formulation in Equation 1, a HyperPINN could consist of

$$\theta_{\hat{u}} = f_h(\lambda; \theta_h), \ \hat{u} = \hat{u}(t, x; \theta_{\hat{u}}). \tag{2}$$

Here $f_h$ is the hypernetwork with learneable parameters $\theta_h$, and $\hat{u}$ is the approximate solution. These parameters can be trained using a standard supervised loss, but also physics-informed losses such as the ones in Equations 6 or 9. Appendix A.4 contains a schematic representation of the HyperPINN.

Table 1: Comparison of HyperPINN and baselines on the parameterized Burgers' PDE.

| Model | Mean squared error | Model size | Evaluation time |
|---|---|---|---|
| Small PINN | $3.0 \cdot 10^{-4}$ | 401 parameters | $92\mu s$ |
| Large PINN | $2.3 \cdot 10^{-5}$ | 9665 parameters | $158\mu s$ |
| HyperPINN | $1.9 \cdot 10^{-5}$ | Main: 393 parameters
Hyper: 9385 parameters | Main: $86\mu s$
Hyper: $158\mu s$ |

## 3 Experiments

We evaluate the application of HyperPINNs to both an example PDE problem, the 1D Burgers' PDE, and an example ODE problem, the Lorenz system, using both a PINN and a multistep neural network.

### 3.1 1D Burgers' equation

The 1D Burgers' PDE, with solution $u(t, x)$, a function of time $t$ and spatial coordinate $x$, is given by

$$\frac{\partial u}{\partial t} + u \frac{\partial u}{\partial x} - \nu \frac{\partial^2 u}{\partial x^2} = 0. \tag{3}$$

This equation has a viscosity parameter $\nu$, which can cause the underlying solution $u$ to exhibit different behavior depending on its value. For low values, such as $\nu = 0.001$, the solution develops a characteristic shock. For higher values, such as $\nu = 0.1$, the solution is smooth. In order to test the ability of the PINN and the HyperPINN to learn parameterized systems, we performed this experiment by having the parameter $\nu$ vary in the range $[0.001, 0.1]$. For training, a dataset with 100 random points from the initial and boundary conditions was created, with the additional collocation points, used in the physics-informed loss, sampled randomly. For more details refer to Appendix A.5.

We evaluate the HyperPINN and compare it to two PINN baselines. For the HyperPINN, the hypernetwork takes as input the parameter $\nu$ and outputs the parameters for a main neural network, which takes as input the space-time coordinates and approximates the solution function $u$. For the standard PINN baselines, a single network takes as input both the parameter $\nu$ and the space time coordinates, and outputs the approximate solution. The summary of the results are shown in Table 1.

The HyperPINN consists of a main network with 6 fully-connected (FC) hidden layers of size 8 (393 parameters) and a hypernetwork with 3 FC hidden layers of size 32 followed by 1 hidden layer of size 16 (9385 parameters). The first baseline, the small PINN baseline, consists of a FC network of approximately the same size as the main network in the HyperPINN, with 6 hidden layers of size 8 (401 parameters). This baseline serves as a comparison point to the main network, which is the one that is evaluated multiple times after the hypernetwork is evaluated once to generate the main network's weights. Qualitatively, we can see in Figure 1 that the HyperPINN learns to approximate the solution function even at a parameter value containing a shock. The small PINN baseline is not able to learn to properly approximate the solution, lacking capacity to fully learn the parameter space.

The second baseline, the large PINN baseline, consists of a FC network of approximately the same size as the full hyper and main networks combined in the HyperPINN, with 10 hidden layers of size 32 (9665 parameters). This baseline serves as a comparison point with the same full capacity as the HyperPINN. It is able to achieve results close to the HyperPINN, but it has a much larger size. As a consequence, the large PINN has a runtime of $158\mu s$ for performing a single prediction, whereas the main network in the HyperPINN has a runtime of $86\mu s$ for performing the same prediction, which is faster than even the small PINN. That is, for a similar number of parameters, the HyperPINN is able to surpass the performance of a large network while keeping the main network as efficient as a small one, by offloading the task of learning the parameter space to the hypernetwork. All times were measured on an NVIDIA Tesla V100 GPU.

### 3.2 Lorenz system

To show this method works with diverse physics-informed approaches, in this experiment we use a multistep neural network to learn the time-derivative of the Lorenz system right-hand side in Equation

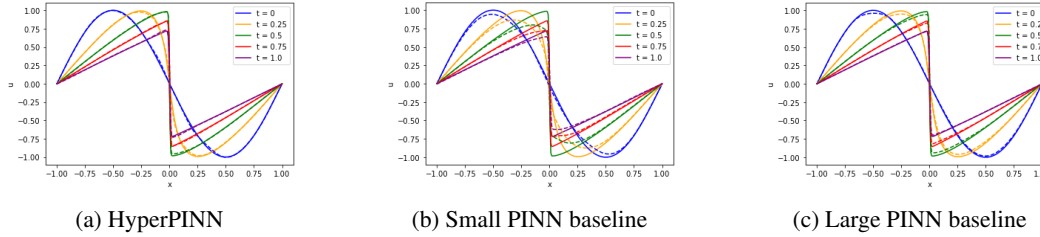

| (a) HyperPINN | (b) Small PINN baseline | (c) Large PINN baseline |

Figure 1: Results for the parameterized 1D Burgers' PDE at a series of time points for $\nu = 0.003$. Solid lines represent the ground truth simulation, and dashed lines represent the model predictions.

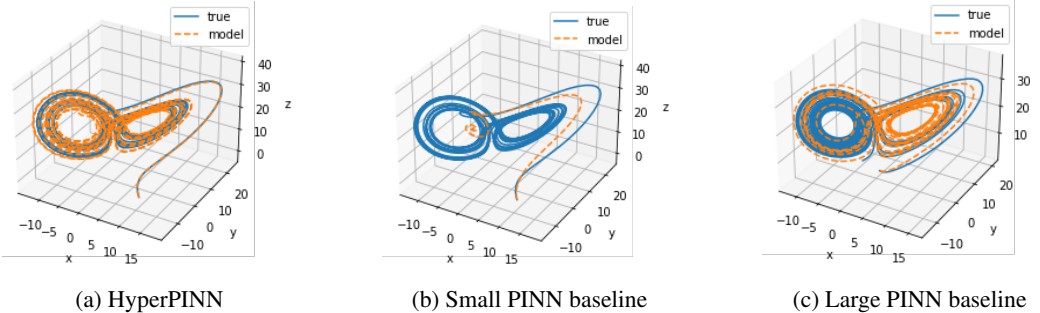

| (a) HyperPINN | (b) Small PINN baseline | (c) Large PINN baseline |

Figure 2: Results for the Lorenz ODE with $\sigma = 10$, $\beta = 5/3$ and $\rho = 21.7$. Solid lines represent the ground truth trajectory, and dashed lines represent the trajectory integrated from model predictions.

4), instead of the solution function. The Lorenz ODE with solution $(x, y, z)$ is given by

$$\left( \frac{dx}{dt}, \frac{dy}{dt}, \frac{dz}{dt} \right) = \left( \sigma(y - x),\ x(\rho - z) - y,\ xy - \beta z \right) \tag{4}$$

This equation has parameters $(\sigma, \beta, \rho)$. In order to test the ability of the PINN and the HyperPINN to learn parameterized systems, we performed this experiment with $\sigma = 10$, $\rho$ varying in the range $[0, 28]$, $\beta$ in $[2/3, 8/3]$, and diverse initial conditions in $[-10, 10]^3$. These values cause the underlying solution to exhibit diverse behavior, including the chaotic attractor. Evaluation is performed using parameter and initial condition values not seen in training. For more details refer to Appendix A.6.

We compare the HyperPINN with two baselines, a small and a large multistep neural network. The HyperPINN consists of a main FC network with one hidden layer of size 16 (115 parameters) and a FC hypernetwork with 2 hidden layers of size 16 and 8 (1406 parameters). The small baseline consists of a FC network with one hidden layer of size 16 (214 parameters). The large baseline consists of a FC network with one hidden layer of size 256 (3334 parameters). In the HyperPINN, the hypernetwork takes as input the parameters $(\sigma, \beta, \rho)$ and outputs the parameters for a main network, which takes as input the state $(x, y, z)$ and outputs the time-derivative $(\dot{x}, \dot{y}, \dot{z})$. In the baselines, a single network takes as input both the parameters and the state, and outputs the time-derivative.

Note that when performing integration using the learned model, it is expected that the system will deviate from the ground truth. Given the chaotic nature of the system, even minuscule deviations from the true time-derivative compound over time when performing integration, causing large errors even when the correct qualitative behavior is captured. Nevertheless, when compared to the baselines, the HyperPINN achieves lower error in predicting the trajectories of the system. While the HyperPINN achieves an aggregate squared error over all test trajectories of 17.5, the small baseline has an error of 39.4, and the large baseline has an error of 20.6. Importantly, when analyzing the results qualitatively, we can see that the HyperPINN achieves this lower error by more accurately capturing the qualitative behavior of the system at different parameter values. In comparison, the baselines frequently mistake one type of behavior for another, as exemplified in Figure 2.

Particularly, the HyperPINN achieves better performance than the small baseline with a similarly sized main network. In comparison to the large baseline, the HyperPINN is able to approximate the solution at diverse parameters and initial conditions while using a much smaller sized main network, which is repeatedly evaluated for integration at test time.

## 4  Conclusion

We have presented an approach to learning parameterized families of differential equations that separates the learning task into two parts. By using a separate hypernetwork to learn the parameter space, the main network that approximates the differential equation solution can remain small. This approach allows physics-informed neural networks to scale up more efficiently to learn parameterized families of differential equations at once, instead of having to be re-trained for every parameterization, without losing their compactness and efficiency. In this work, we demonstrated this approach on simple ODE and PDE problems. In the future, we intend to apply this method to more complex domains, such as higher dimensional problems and more diverse initial or boundary conditions.

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

# A Appendix

## A.1 Differential equations

A general differential equation is given by

$$\mathcal{N}[t, x, u(t, x); \lambda] = 0, \ t \in [0, T], \ x \in \Omega, \tag{5}$$

where $\mathcal{N}[\cdot; \lambda]$ is an arbitrary (possibly non-linear) differential operator, which can contain time and space derivatives, and is parameterized by some list of parameters $\lambda \in \mathbb{R}^d$. Here, $t$ is the time variable ranging up to time $T$, $x$ is the D-dimensional spatial variable in some domain $\Omega \subseteq \mathbb{R}^D$, with boundary $\partial\Omega$, and $u(t, x)$ is the solution function to the differential equation. For a solution to be defined, initial and boundary conditions need to be provided. These can assume different forms, but in general initial conditions define $u(0, x)$ for $x \in \Omega$, and boundary conditions define $u(t, x)$ for $x \in \partial\Omega$ and $t \in [0, T]$.

For a concrete example of this formulation, refer to Equation 3, in which $\mathcal{N}$ is a non-linear operator containing first and second derivatives that defines the left-hand side of the equation, and $\lambda = \nu$. The initial and boundary conditions are given in Equation 10.

## A.2 Physics-informed neural networks

Physics-informed neural networks [9] are a method for approximating the solution to differential equations using neural networks (NNs). In this method, a neural network $\hat{u}(t, x; \theta)$, with learned parameters $\theta$, is trained to approximate the solution function $u$ to the differential equations.

Importantly, PINNs employ not only a standard "supervised" data loss, but also a physics-informed loss, which consists of the differential equation residual defined by $\mathcal{N}$. Thus, for a given optimization hyper-parameter $\alpha$, the training loss consists of

$$
\begin{aligned}
L(\theta) &= L_{\text{data}}(\theta) + \alpha L_{\text{physics}}(\theta), \\
L_{\text{data}}(\theta) &= \sum_{(t_i, x_i, u_i) \in \mathcal{D}} [\hat{u}(t_i, x_i; \theta) - u_i]^2, \\
L_{\text{physics}}(\theta) &= \sum_{(t_c, x_c) \in \mathcal{C}} \mathcal{N}[t_c, x_c, \hat{u}(t_c, x_c; \theta); \lambda]^2,
\end{aligned}
\tag{6}
$$

where $\mathcal{D}$ is a dataset containing ground-truth values for $u$ (e.g., from simulation data) at points $(t_i, x_i)$, and $\mathcal{C}$ is a set of collocation points at which to evaluate the differential equation residual (which does not require ground-truth solution data). While the data in $\mathcal{D}$ can be used to enforce the initial and boundary conditions, the physics-informed loss term regularizes the search space, penalizing functions $\hat{u}$ that do conform to the differential equations, reducing the need for simulation data.

## A.3 Multistep neural networks

Multistep neural networks [8] are a related method, in which a neural network model is used to approximate the time-derivative of a dynamical system. Instead of using the differential equation residuals, the loss for the multistep neural network is derived directly from the formulation for traditional multistep methods. That is, for a general dynamical system

$$\frac{d}{dt}x(t) = f(x(t)), \tag{7}$$

where $f$ is some arbitrary (linear or non-linear) function of $x \in \mathbb{R}^D$, a two-step linear multistep method, with timestep $\Delta t$, gives us the relation

$$x_n = x_{n-1} + \frac{1}{2}\Delta t(f(x_n) + f(x_{n-1})). \tag{8}$$

If we want to train a neural network $\hat{f}(\cdot; \theta)$ to approximate the time derivative $f$, this relation can then be used to define a residual loss over a dataset of points $x_n$ sampled from a given trajectory

$$L(\theta) = \sum_{(x_n, x_{n-1}) \in \mathcal{D}} [x_n - x_{n-1} - \frac{1}{2}\Delta t(\hat{f}(x_n; \theta) + \hat{f}(x_{n-1}; \theta))]^2. \tag{9}$$

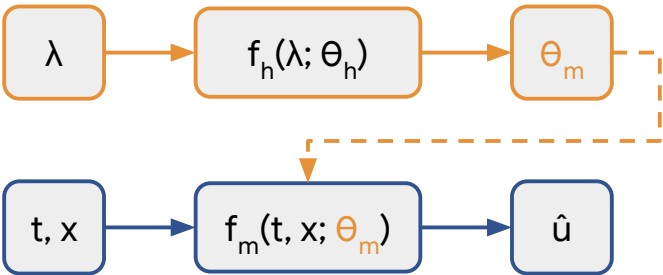

Figure 3: Diagram for the HyperPINN. The hypernetwork, represented in orange, takes as input the parameterization and outputs the parameters for the main network. The main network, represented in blue, takes as input a space-time coordinate and outputs its prediction using the parameters provided by the hypernetwork.

## A.4 HyperPINN

Figure 3 contains a schematic representation of the hyper and main networks that compose the HyperPINN. The hypernetwork takes as input the parameterization and outputs the parameters for the main network. The main network takes as input a space-time coordinate and outputs its prediction using the parameters provided by the hypernetwork.

For the case Burgers' PDE, $\lambda$ corresponds to the parameter $\nu$ in the PDE, and the main network outputs and approximation of the solution $u$. For the case of the Lorenz ODE, $\lambda$ corresponds to the parameters $(\sigma, \beta, \rho)$. When using the multistep neural network approach, the main network takes as input the spatial coordinates $(x, y, z)$ and outputs the time-derivative $(\dot{x}, \dot{y}, \dot{z})$.

Here, $\theta_h$ are the learned parameters. These are trained using a loss on the predictions of the main network, which can then be optimized with gradient-based methods using conventional automatic differentiation packages, since all operations are differentiable. If employing a stochastic gradient descent method, batches of randomly sampled parameter-input pairs can be used. The loss can consist of a regular supervised loss, with a ground truth for the main network prediction for each given parameterization and inputs. Moreover, physics-informed losses can also be used, such as the PINN loss or the multistep loss defined above, in order to make the training more data-efficient.

## A.5 1D Burgers' PDE

For the 1D Burgers' experiment we use as our time domain the range $[0, 1]$ and as our spatial domain the range $[-1, 1]$. We utilized a sinusoidal initial condition and a Dirichlet boundary condition given by

$$
\begin{aligned}
u(0, x) &= -\sin(\pi x), \\
u(t, -1) &= u(t, 1) = 0.
\end{aligned}
\tag{10}
$$

A dataset of size 100 was generated with 50 points sampled randomly uniformly at different positions from the initial timestep ($t = 0$) and 50 points sampled randomly uniformly at diverse timesteps from the boundary ($x = -1$ or $x = 1$). Since these values do not change with the parameterization, values of the parameter $\nu$ are sampled randomly uniformly from the range $[0.001, 0.1]$ at training time to form the training data points. The collocation points are sampled randomly uniformly from the time-space domain, with accompanying values for $\nu$ also sampled randomly uniformly.

Example solutions for different parameter values are shown in Figure 4.

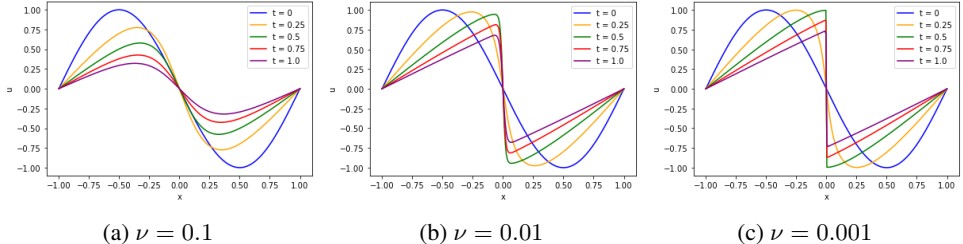

(a) $\nu = 0.1$          (b) $\nu = 0.01$          (c) $\nu = 0.001$

Figure 4: Solutions from the 1D Burgers' PDE, displaying diverse behavior for different parameter values.

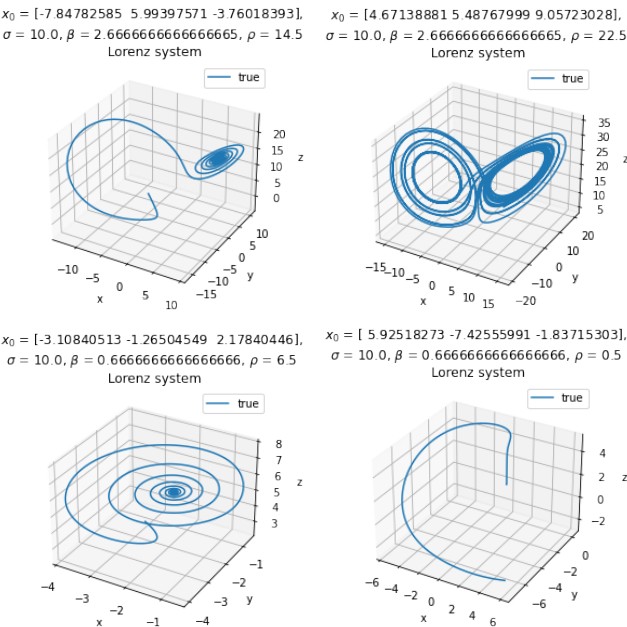

Figure 5: Trajectories from the Lorenz system, displaying diverse behavior for different initial conditions and parameter values.

## A.6   Lorenz system

For the Lorenz experiment, we sampled 100 different initial conditions randomly uniformly from $[-10, 10]^3$ for each of 30 different parameter values in the ranges described in Section 3.2.

We computed trajectories of 25s duration with $\Delta t = 0.01$ using the Runge-Kutta 45 integrator. Evaluation is performed using a separate set of 100 trajectories, sampled from the same range of parameters and initial conditions, but using different values from the training set.

Example trajectories for different initial conditions and parameter values are shown in Figure 5.

