# OpenReview forum: "HyperPINN: Learning parameterized differential equations with physics-informed hypernetworks"
_NeurIPS.cc/2021/Workshop/DLDE — DLDE Workshop -- NeurIPS 2021 Poster_

### Official Review · Reviewer_EGbv · 2021-10-03
**Well written paper with a good technical contribution**

**Confidence:** 3

**Review:**

This paper proposes to meta-learn the solution to PDEs and ODEs with different parameterisations using hypernets. The paper is very well written, and the technical contribution has merit and the results demonstrate this.

I have some small questions and suggestions:

- When considering the evaluation time, should the hypernet evaluation time be included because it is required to generate the main network?

- A paragraph considering how the hypernets can be viewed as PINNs with more meaningful structure could improve the paper. This can be seen in equation 2 which can be rewritten as, $\hat{u} = \hat{u}(t, x ; f_h(\lambda, \theta_h)) = \hat{u}_{new}(t, x, \lambda, \theta_h)$. The PINNs in the experiments attempt to model this directly with one neural network, whereas the hypernets introduce additional meaningful structure by having a composition of a hyper network and a main network.

**Score:**

4: Very good paper

---

### Official Review · Reviewer_QuY7 · 2021-10-12
**Well-written with good results**

**Confidence:** 3

**Review:**

### Summary

The paper proposes a parametrization-aware variant of physics-informed neural networks by introducing an hyper-network outputting the parameters of the the differential equation's approximate solution (i.e. the *main* neural net). The paper is well-written and the experiments show a clear improvement w.r.t. the proposed baselines.

### Minor Comments

I wonder whether it would be possible to directly learn a parametrization-aware solution of the ODE/PDE by feeding the physical parameters $\lambda$ directly into the main neural network, i.e.
$$
    t, x, \lambda ; \theta \mapsto \hat u(t, x, \lambda; \theta)
$$
In my understanding, this approach could theoretically increase even further the parameter efficiency (and inference time) of the model since we do not have to rely on a rather costly hyper-network.

**Score:**

3: Good paper

---

### Official Review · Reviewer_pbov · 2021-10-12
**Meta-learning for different DE parameterizations**

**Confidence:** 3

**Review:**

This paper proposes the use of a meta-learning method to learn different parameterizations of differential equations. The proposed HyperPINN network receives the parameters of the DE and outputs the parameters of the main network which is the approximate solution to the DE. The paper is well-written and the results show improvement over the baseline methods.

Questions:
How are HyperPINNs trained? I think it would be good to add a brief description.

**Score:**

3: Good paper

---

### Decision · Program_Chairs · 2021-10-16

**Decision:**

Accept (Spotlight)

**Comment:**

All reviewers have recommended acceptance.